# Application of High-Photoelasticity Polyurethane to Tactile Sensor for Robot Hands

**DOI:** 10.3390/polym14235057

**Published:** 2022-11-22

**Authors:** Masahiko Mitsuzuka, Jun Takarada, Ikuo Kawahara, Ryota Morimoto, Zhongkui Wang, Sadao Kawamura, Yoshiro Tajitsu

**Affiliations:** 1Research Organization of Science and Technology, Ritsumeikan University, Kusatsu 525-8577, Japan; 2Electrical Engineering Department, Graduate School of Science and Engineering, Kansai University, Suita 564-8680, Japan; 3Cloud Robotics Laboratory, Department of Robotics, Ritsumeikan University, Kusatsu 525-8577, Japan; 4Ritsumeikan Global Innovation Research Organization, Ritsumeikan University, Kusatsu 525-8577, Japan

**Keywords:** photoelasticity, tactile sensor, tangential force

## Abstract

We developed a tactile sensor for robot hands that can measure normal force (F_Z_) and tangential forces (F_X_ and F_Y_) using photoelasticity. This tactile sensor has three photodiodes and three light-emitting diode (LED) white light sources. The sensor is composed of multiple elastic materials, including a highly photoelastic polyurethane sheet, and the sensor can detect both normal and tangential forces through the deformation, ben sding, twisting, and extension of the elastic materials. The force detection utilizes the light scattering resulting from birefringence.

## 1. Introduction

### 1.1. History

Various methods have been studied to detect the stress inside polymers using the birefringence of light [1,2,3,4,5,6,7,8,9,10,11,12,13,14,15,16,17,18,19,20,21,22,23,24,25,26,27,28,29]. A tactile sensor for stress detection using photoelastic, transparent, flexible, and robust polyurethane is expected to realize a highly sensitive robot hand tactile sensor that imitates the tactile sensation of a human finger, but the performance in grasping and lifting objects has not yet been fully verified [18,25].

Thus, we have found that polyurethane with a specific structure exhibits high photoelasticity, as we constructed a flexible force sensor using this polyurethane, as published previously [27]. Figure 1 shows the mechanism for detecting the force perpendicular to the polyurethane sheet; in other words, the normal force. The birefringence generated inside a polymer scatters the incident light [30,31,32,33]. The mechanism of force-induced birefringence is explained in Appendix B [27].

### 1.2. Purpose of This Study

Our previously developed force sensor could not measure the force horizontal to the polyurethane sheet; in other words, the tangential force. Therefore, in this study, we devised a tactile sensor that can measure both normal and tangential forces.

One of the envisioned applications of this tactile sensor is a two-finger manipulator for robots [26]. Figure 1 shows the two-finger manipulator used in this study.

In this manipulator, the rotating shaft of a motor is connected to a pinion gear, and when the pinion gear rotates, the two opposing rack gears move parallel to each other, in opposite directions.

Two fingers are attached, one to each of these two rack gears, and the two fingers open and close as the rack gears move in parallel. The finger spacing is controlled by the rotation angle of the pinion gear.

Figure 2 shows the manipulator grasping and lifting a soft bottle on a table. In the initial state, the bottle is stationary on the floor and the fingers of the manipulator do not touch the bottle. In the grasping stage, the manipulator’s fingers close and grasp the bottle. In this stage, the tactile sensor measures the normal force. In the lifting stage, the manipulator lifts the bottle, while maintaining the distance between the fingers. In this stage, the tactile sensor measures the tangential force due to the weight of the bottle. If the tangential force can be measured, the weight of the bottle can be determined. If the bottle slips during lifting, the tangential force is no longer detected, immediately confirming that the robot failed to grasp the bottle.

A robot also typically uses a camera and artificial intelligence image analysis to determine the position of grasped objects on a table. However, due to image analysis limitations, the robot may misidentify the position of the object. By measuring the tangential force, the robot can confirm that the bottle was grasped securely. This technology is very useful for systems that use mobile robotic dollies to serve food and drink, and to clear dishes [34,35,36]. The mobile robot may incorrectly measure the position of the tableware. This sensor helps the robot to instantly determine whether the tableware is fully grasped or not.

The goal of our research was to construct a low-cost grasping system, using a commercially available robot hand, and to apply it to a mobile robotic dolly system. The strain gauges used in traditional force sensors require expensive electronics to obtain a signal. In this study, we develop a tactile sensor that can detect signals with inexpensive measurement equipment.

Furthermore, in order to use a commercially available two-finger robot hand, the tactile sensor needs, not only a sensing function but also a cushioning function that does not lose the shape of the object when grasping a soft and fragile object [27]. This is because the ordinary commercially available robot hands do not have a function to control the grasping force of the hand by feeding back the grasping force detected by the tactile sensor. The tactile sensor’s cushioning function avoids applying a sudden force to the object.

This technology also has potential applications as a wearable human–machine interface [37,38,39,40,41,42,43,44].

## 2. Experimental Section

### 2.1. Outline

Figure 2 shows the structure of the tactile sensor in this work. The basic structure is almost the same as that of the force sensor in our previous paper [27], but the number of photodiodes is different. Three photodiodes, which detect scattered light, are embedded in a soft, black, 3 mm thick base rubber sheet with a Young’s modulus of 3.4 MPa. A silicone sheet of 0.2 mm thickness is laminated on the base rubber. The role of the silicone sheet was described in the previous paper [27].

Furthermore, a 2 mm thick polyurethane sheet and a 0.2 mm thick silicon sheet are sequentially laminated on the upper surface of the laminate. The Young’s modulus of the polyurethane sheet is 4.7 MPa. The layers are glued together with an elastic adhesive [27], and then a top cover made of soft black rubber with a thickness of 1 mm and a Young’s modulus of 3.4 MPa is glued on top of these layers. Then, the whole laminate is adhered to a support material made of hard resin. The support material is concave in the center, and the polyurethane sheet flexes into a convex shape when the cover material is pressured from above. The polyurethane used in this study has the sample name AR-2, whose composition and properties are described in detail elsewhere [27,28].

The entire sensor has the flexibility of rubber. The grasping surface of the sensor can be deformed into a concave shape of 1 mm or more in the normal direction, so that an object can be grasped without applying a sudden force. As a result, damage to the grasping object is avoided.

In this paper, the direction of force is indicated by the coordinate axes fixed to the tactile sensor, with the Cartesian *x*-axis parallel to the short side of the polyurethane sheet, the *y*-axis parallel to the long side of the polyurethane sheet, and the *z*-axis perpendicular to the polyurethane sheet surface.

The major difference from the force sensor in the previous paper [27] is that there are three photodiodes arranged in a triangular shape, as shown in Figure 3. Photodiodes PD1 and PD2 are placed parallel to the X-direction and the same length apart from the center of the sensor in opposite directions, while photodiode PD3 is shifted from the center of the sensor in the Y-direction.

By arranging the photodiodes in this way, the normal force (F_Z_) can be obtained from the average value of the outputs of PD1 and PD2, the tangential force (F_X_) in the X-direction is obtained from the difference between the outputs of PD1 and PD2, and the tangential force (F_Y_) in the Y-direction is obtained from the output of PD3.

Three LEDs are used to irradiate the interior of the polyurethane sheet with light of uniform intensity.

This sensor utilizes the birefringence caused by the strain generated inside the polyurethane sheet. Figure 4a schematically shows the intensity of birefringence generated inside the polyurethane sheet when a normal force (F_Z_) is applied to the sensor. The polyurethane sheet is flexed by the normal force, creating birefringence in the center of the sheet.

Since the birefringence intensities generated near photodiodes PD1 and PD2 are almost equal, it is reasonable to assume that the average value of output of PD1 and PD2 represents the intensity of birefringence generated by the normal force.

Figure 4b conceptually shows the twisting of the polyurethane sheet when a tangential force F_X_ is applied to the sensor simultaneously with the normal force F_Z_, and the resulting change in the position at which birefringence occurs. It was also confirmed by the simulation, using a finite element method, that such a twisting is caused by the tangential force F_X_.

When the twisting occurs, the displacement of the polyurethane sheet in the Z-direction decreases in the vicinity of photodiode PD1 and increases in the vicinity of photodiode PD2. As a result, the center of the birefringence region moves away from photodiode PD1 and approaches photodiode PD2. Such changes are detected as a decrease in output of PD1 and an increase in output of PD2.

Figure 4c conceptually illustrates the extension of the polyurethane sheet when a tangential force F_Y_ is applied to the sensor simultaneously with the normal force F_Z_ and the resulting change in the position of the birefringence. This extension, caused by the tangential force F_Y_, was also confirmed by finite element simulation.

When extension occurs, the displacement of the polyurethane sheet in the Z-direction decreases in the vicinity of the photodiode PD3, and the center of the region of birefringence moves away from PD3. Such a change in the central position of birefringence is detected as a decrease in output of the PD3.

This tactile sensor is characterized by its simplicity, in that the normal and tangential forces applied to the grasping object can be obtained from the output of only three photodiodes without any complicated amplification circuits.

Figure 3 shows the circuits used for detecting the outputs of the photodiodes. Each photodiode is connected in series with a DC power supply and a resistor.

The current flowing out of each photodiode is detected by measuring the potential difference between the two ends of the resistor. From the values of current (j_n_) and resistance (R_n_), the output V_n_ is obtained using
V_n_ = j_n_ × R_n_(1)
where *n* = 1, 2, and 3 correspond to PD1, PD2, and PD3.

Hamamatsu Photonics S9066-211SB photodiodes are used, and these photodiodes have a built-in amplification circuit that amplifies the generated photocurrent by a factor of several thousand and outputs it [45].

Next, we describe how to adjust the resistance of the circuit. When the light-emitting diode (LED) is turned on with no stress applied to the tactile sensor, part of the incident light is detected by the photodiode. The resistor value R_n_ is adjusted so that the output V_n_ (*n* = 1, 2, or 3) becomes 2.0 V. Figure 5a is an oscillogram showing the outputs of the three photodiodes when an object is grasped and lifted. Figure 5b shows the effective output as V^ef^_n_ (*n* = 1, 2, 3), which is obtained by subtracting the output before grasping (about 2.0 V) from each output.

### 2.2. Measurement of Normal Force F_Z_

Figure 6a shows how the normal force F_Z_ is measured using a force gauge. V_12Z_ is the mean of V^ef^_1_ and V^ef^_2_ when the normal force is F_Z_:V_12Z_ = (V^ef^_1_ + V^ef^_2_)/2(2)

Figure 6b is a graph showing the relationship between V_12Z_ and F_Z_. F_Z_ can be expressed as a quadratic function of V_12Z_:F_Z_ = C1 × V_12Z_ + C2 × (V_12Z_)^2^(3)
where C1 and C2 are constants.

Equation (4) is obtained by finding the constants that best match the values calculated using Equation (3) and the measured values:F_Z_ = 25.2 × V_12Z_ − 13.6 × (V_12Z_)^2^(4)

Figure 7a is a photograph of a robot hand grasping a cuboid object; the object is sandwiched between the left and right tactile sensors. The normal force is measured with the left tactile sensor. The right tactile sensor, which has the same elastic properties as the left tactile sensor, does not measure the force.

The object rests on a table that can move freely horizontally. Therefore, when the object is grasped from both sides, the normal forces F_Z_ applied to the tactile sensors on both sides are opposite in direction but of the same in magnitude. The normal force F_Z_ is obtained from the measured value of V_12Z_ and using Equation (4).

### 2.3. Measurement of Tangential Force F_X_

Figure 7b is a photograph of the robot hand lifting a cuboid parallel to the sensor’s *x*-axis and perpendicular to the table.

The grasping surface of the sensor was made with 3D printing technology and is made of flexible, flat rubber, but is not adhesive.

Cuboids were used in these experiments because the center of gravity of cuboids is at the center and the sides of the cuboid are vertical, which simplified the analysis of tangential forces. The cuboids were made of aluminum or iron, and their surfaces were smooth. The experiments were conducted to the extent that slippage did not occur.

A tangential force F_X_ was applied to the tactile sensor in the X-direction; the *x*-axis of the sensor was vertical. Figure 8a is an oscillogram showing the time variation of V^ef^_1_, which decreases by ΔV_1X_ when transitioning from the grasping stage to the lifting stage (Appendix A).

Figure 8b is an oscillogram showing the time variation of V^ef^_2_, which increases by ΔV_2X_ when transitioning from the grasping stage to the lifting stage.

The sum of ΔV_1X_ and ΔV_2X_ is ΔV_12X_:ΔV_12X_ = ΔV_1X_ + ΔV_2X_(5)

The outputs in Figure 8 rapidly decreases or increases when the object is lifted. This is the result of immediate twisting of the polyurethane sheet. The relationship between the sheet twisting and the output change is explained in Figure 4b.

### 2.4. Measurement of Tangential Force F_Y_

Figure 9 is a photograph of the object lifted parallel to the *y*-axis of the sensor and perpendicular to the table. A tangential force F_Y_ is applied to the tactile sensor in the Y-direction. Figure 10a is an oscillogram showing the time variations of V^ef^
_1_, V^ef^
_2_, and V^ef^
_3_. Figure 10b shows a rescaled graph of V^ef^
_3_. V^ef^
_3_ decreases by ΔV_3Y_ from the grasping stage to the lifting stage.

The output in Figure 10b decreases rapidly when the object is lifted. This is a result of the force applied to the polyurethane sheet, causing the sheet to immediately extend in the Y-direction (Appendix A). The relationship between the sheet extension and the output reduction is explained in Figure 4c.

## 3. Results and Discussion

### 3.1. Analysis Result of Tangential Force F_X_

For the cuboid-lifting test shown in Figure 7, seven metal cuboids with different weights, ranging from 105 g to 589 g, were used.

The tangential force F_X_ is given by the following formula:F_X_ = *m* × *g*/2(6)
here, *m* is the weight of the cuboid and the g is gravitational acceleration.

The lifting test was conducted for grasping forces F_Z_ of 9.6 N, 6.4 N, and 4.1 N. The results are shown in Figure 11. Each plot is the average of three measurements.

All measurements were on a straight line through the origin. The slope of the straight line did not change even when the normal force F_Z_ was changed.

Therefore, when the value of ΔV_12X_ is obtained from the output of the tactile sensor, the tangential force F_X_ can be obtained using the following formula, independently of the normal force:F_X_ = 25.9 × ΔV_12X_(7)

### 3.2. Analysis Result of Tangential Force F_Y_

For the cuboid-lifting test shown in Figure 9, six metal cuboids, with weights ranging from 105 g to 589 g, were used.

Denoting the weight of the object by *m* and gravitational acceleration by *g*, the tangential force F_Y_ is obtained using the following formula:F_Y_ = *m* × *g*/2(8)

Lifting tests were carried out for normal forces F_Z_ of 9.7 N, 8.3 N, and 6.7 N. The results are shown in Figure 12. Each plot is the average of three measurements.

The measurement results were plot on different straight lines for each F_Z_, but the slopes of the straight lines were the same. 

Therefore, the relationship between ΔV_3Y_, normal force F_Z_, and tangential force F_Y_ is expressed by the following equation:ΔV_3Y_ = C3 + C4 × F_Z_ + C5 × F_Y_(9)
where C3, C4, and C5 are constants.

Since the normal force F_Z_ and the tangential force F_Y_ are known, the constants that give the best agreement between the values of ΔV_3Y_ calculated using Equation (9) and the measured values of ΔV_3Y_ yield
ΔV_3Y_ = −0.0529 + 0.0118 × F_Z_ + 0.0443 × F_Y_(10)

The dotted line in Figure 12 shows the result of Equation (10). From this equation, the tangential force F_Y_ can be obtained using the values of the normal force F_Z_ and the measured value of ΔV_3Y_.

### 3.3. Accuracy of Measured Tangential Force

The iron cuboids weighing 211 g, 297 g, 394 g, and 493 g were each grasped and lifted nine times, with the sensor’s *x*-axis vertical. Figure 13 shows the relationship between the actual cuboid weight and the weight calculated from the sensor output ΔV_12X_ using Equations (6) and (7). The circles in the figure indicate the average values of nine measurements, and the horizontal bars indicate the maximum and minimum values. The error between the average value of nine measurements and the actual cuboid weight was less than 4% of the actual weight. The standard deviations of nine measurements were 15.6 g for the 211 g cuboid and 42.0 g for the 493 g cuboid.

### 3.4. Discussion

Figure 4a schematically shows the intensity of birefringence generated inside a polyurethane sheet when a normal force F_Z_ is applied to the sensor. The average values of V^ef^_1_ and V^ef^_2_ shown in Figure 5 are expected to represent the strength of birefringence due to normal force. Equation (3) shows the relationship between the output V_12Z_ defined by Equation (2) and the normal force. Below, we use Equation (3) to evaluate the normal force.

When the polyurethane sheet is twisted, as shown in Figure 4b, the Z-direction displacement of the polyurethane sheet decreases near the photodiode PD1 and increases near the photodiode PD2. Such changes are detected as a decrease in V^ef^_1_ and an increase in V^ef^_2_. This change is expressed as an increase in ΔV_12X_, defined by Equation (5).

As shown in Figure 11, ΔV_12X_ and F_X_ are proportional, as shown in Equation (7). It can be seen that this relationship is independent of the magnitude of the normal force.

When the polyurethane sheet is stretched, as shown in Figure 4c, the Z-direction displacement of the polyurethane sheet decreases in the vicinity of the photodiode PD3, and the center of the birefringent region moves away from PD3. Such a change in the center position of birefringence was detected as a decrease in V^ef^_3_, as shown in Figure 10.

The tangential force along the *y*-axis is expressed as a function of V_3Y_ and the normal force, as shown in Equation (10).

As shown in Figure 13, the tangential force measurement accuracy of this sensor is limited. When a 211g cuboid was weighed nine times, the average weight (μ) was 208 g and the standard deviation (σ) was 15.6 g. Assuming that the measurement results follow a normal distribution, one measurement result will fall within the range μ ± 3σ with a probability of 99.7%. It is estimated that re-weighing the 211 g cuboid would give a value in the range of 161 g to 255 g with a probability of more than 99%. Therefore, the measurement accuracy of the tangential force of this sensor is sufficient to confirm that an object, such as tableware or a soft bottle, has been lifted.

Interestingly, the measurement error tended to increase as the weight of the cuboid increased. This phenomenon can be considered as follows:

The heavier the cuboid, the greater the deformation of the rubber of the top cover of the tactile sensor, causing a partial slipping phenomenon at the contact interface between the rubber and the cuboid; and the degree of twisting of the polyurethane sheet may vary for each measurement. Clarifying the relationship between the weight of the cuboid and this error is an issue for the future.

## 4. Conclusions

The sensor we developed previously could only measure the normal force. However, this sensor, which has three LEDs and three photodiodes arranged at specific positions, can measure, not only the normal force in the *z*-axis, but also the *x*-axis tangential force and the *y*-axis tangential force.

The magnitude of the normal force F_Z_ applied to the tactile sensor in the grasping stage can be estimated from the mean value V_12Z_ of the effective outputs of photodiodes PD1 and PD2:F_Z_ = 25.2 × V_12Z_ − 13.6 × (V_12Z_)^2^(11)

The tangential force F_X_ to the tactile sensor when the normal force F_Z_ is applied to the tactile sensor can be obtained from the following equation:F_X_ = 25.9 × ΔV_12X_(12)
where ΔV_12X_ is the sum of the effective decrease in output ΔV_1X_ of photodiode PD1 and the effective increase in output ΔV_2X_ of photodiode PD2.

The tangential force F_Y_ to the tactile sensor can be estimated from the following equation, when the normal force F_Z_ is applied to the tactile sensor:ΔV_3Y_ = −0.0529 + 0.0118 × F_Z_ + 0.0443 × F_Y_(13)
where ΔV_3Y_ is the decrease in the effective output of photodiode PD3.

The measurement accuracy of the tangential force of this sensor is sufficient for confirming that an object, such as tableware or a soft bottle, has been lifted.

Moreover, the strain gauges used in traditional force sensors require low noise, high accuracy, and expensive electronics to deliver the signal to the end user. Therefore, they is not suitable for low-cost grasping systems.

Various grasp sensors have been proposed [46,47,48,49,50,51,52,53,54,55,56,57,58,59,60,61,62,63,64,65], including MEMS sensors [51,52,53,54,55,56,57,58,59,60,61,62,63], some of which were able to measure normal and tangential forces [58,59,60]. However, they had complex structures and required special detection circuits.

Vibro Touch technology [65] can detect the contact between an object and a robot finger with high sensitivity, by detecting attenuation of vibration. This sensor is expected to be cheaper to manufacture than MEMS and strain gauge sensors. Furthermore, since this sensor has a cushion function, it can be attached to a commercially available manipulator.

However, quantification of the grasping force and the tangential force has not yet been demonstrated. In addition, there is the issue of high noise levels, due to the use of an acceleration sensor. Furthermore, vibration attenuation is concerned as it depends on the shape, hardness, density, and elastic modulus of the object, but these have not yet been studied.

The tactile sensor in this study is expected to be low cost because it detects force by converting the photocurrent from the photodiode into a voltage without an amplifier circuit. In addition, the signal noise is low, and almost no processing is required to remove noise. The photodiodes and LEDs used are inexpensive commercial items.

Furthermore, since this tactile sensor has a cushion function, a commercially available robot hand can be used. This is because the ordinary commercially available robot hands do not have a function to control the grasping force of the hand by feeding back the grasping force detected by a tactile sensor. This tactile sensor avoids exerting a strong impact on the object by deforming the grasping surface.

Furthermore, this tactile sensor has a flexible structure, deforms according to the magnitude of the normal force, and follows the shape of the surface of the object. We plan to study the ability to measure normal and tangential forces, even for spherical objects and soft objects.

The development of low-cost grasping systems is desirable, not only in the field of mobile robotic dollies, but also in the medical and nursing care fields [66]. In this system, when a robot hand grasps a cup filled with a drink and carries it to a care recipient, the operator performs grasping operations while observing the cup and the robot hand with the robot’s camera. If the tactile sensor developed in this study is used, the sensor can detect the strength of grasping the cup and the weight when the cup is lifted, so that operation will be easier than when relying only on camera images.

## Data Availability

We complied with all laws and other requirements.

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
