# Peer review of "Application of High-Photoelasticity Polyurethane to Tactile Sensor for Robot Hands"

_polymers, 2022, doi:10.3390/polym14235057_

Round 1
Reviewer 1 Report
Comments
A tactile sensor for robot hands was developed and the grasping and lifting properties of the sensor were tested in the study. The obtained results are interesting and possibly useful for robotic applications.
A video of tactile sensing especially grasping and lifting has to be added to the manuscript.
Reviewer 2 Report
This paper presents a tactile sensor based on a highly photoelastic polyurethane sheet. As a novelty, the authors claim to propose including an additional photodiode, which allows for measuring tangential forces. The proposed photoelastic tactile sensor could have potential interest from researchers. However, there are questions about the presented work that the authors could hopefully address:
- The main idea of sensing is not clear to me. The sensing part should be described appropriately, even if presented in the previous work.
- The introduction is relatively too short and needs a better background description.
- What are the photoelastic polyurethane sheet properties? How do the optical properties change with applied force?
- The evaluation and calibration method seems insufficient. While the normal force measurement was done using a force gauge, the tangential force measurement was done by using objects of different weights and without calibration against other sensors. The latter brings questions regarding how accurate it is and its bandwidth maybe.
- The latter argument puts the significance of the whole paper in question. Is it a significant improvement to be interesting for other researchers? I suggest the authors address these points in the main text before publishing. Or kindly explain what this reviewer missed.
Author Response
Answer to Reviewer 2:
- 1.The main idea of sensing is not clear to me. The sensing part should be described appropriately, even if presented in the previous work.
The property of polymer and the mechanism of force detection are added in the appendix (page 13, line 30-).
- 2.The introduction is relatively too short and needs a better background description.
The final goal of the research was added to "1.2. Purpose of Study" (page 3, line 1-).
- 3.What are the photoelastic polyurethane sheet properties? How do the optical properties change with applied force?
Photoelastic polyurethane has refractive index anisotropy due to force. As a result, part of the incident light is scattered. Details are provided in the appendix (page 13).
- 4.The evaluation and calibration method seems insufficient. While the normal force measurement was done using a force gauge, the tangential force measurement was done by using objects of different weights and without calibration against other sensors. The latter brings questions regarding how accurate it is and its bandwidth maybe.
In "3.3. Accuracy of measured tangential force (page 13)", the tangential force was measured repeatedly and the standard deviation was obtained. We estimated the effect of measurement error on "3.4. Discussion''(page 12, line 9-).
- 5.The latter argument puts the significance of the whole paper in question. Is it a significant improvement to be interesting for other researchers? I suggest the authors address these points in the main text before publishing. Or kindly explain what this reviewer missed.
Our research goal is to develop a grip sensor that can be used with commercially available hands and that can be manufactured at low cost. In order to the robot system to be used in the restaurant industry, hospitals, and nursing homes, it is necessary to assemble the system, including mobile robots, at low cost. I wrote about it in "4. Conclusions"(page 13, line 1-).

Reviewer 3 Report
This manuscript reported tactile sensors for robot hands. The topic is scientific logic. The following issues need to be solved before publication.
1, The introduction is not solid, the authors can cite more papers to support this topic, such as Nano Energy, 103 (2022) 107766.
2, The introduction section needs to condense to make it more logical.
3, What is the novelty in Figure 2 as a tactile sensor
4, Any surface microstructure in Figure 6? How to affect the result in the measurement?
5, Why chose the block object in Figure 8?
6, Why the sudden drop curve in Figure 9b?
7, The conclusion looks bad, please rewrite it.
Round 2
Reviewer 2 Report
This reviewer appreciates the efforts made by the authors to improve the manuscript, especially for extending the sensing and sensor principle background. However, the current structure of the paper feels confusing.
Perhaps, it was not explained well what this reviewer meant by including the information about the sensor principle. While the principle of photoelasticity is well-known and has been discussed in previous works, it is necessary to contain its brief description to highlight the difference and novelty. As it is now, the sensor description is the discussion part, which is very strange. Such a structure makes it difficult to grasp the importance of the third LED and photoelastic layer. I would suggest reorganizing these parts to make the text smooth to read.
Another question that arose after reading the added part 3.3 is the decreased accuracy for heavier objects. It seems counterintuitive. Maybe the authors can explain this in the discussion section.
